# Social Values of Care Robots

**DOI:** 10.3390/ijerph192416657

**Published:** 2022-12-11

**Authors:** Jihwan Kim, Kyongok Park, Hanbyul Ryu

**Affiliations:** 1Department of Economics, Hanyang University ERICA, Ansan-si 15588, Republic of Korea; 2Department of Nursing, Gangneung-Wonju National University, Wonju-si 25457, Republic of Korea

**Keywords:** analytic hierarchy process, care robots, economic values, innovative values, labor values, health values

## Abstract

Care robots have the potential to address the challenge of aging societies, such as labor shortages or the aging workforce. While previous studies have focused mainly on the productivity or workability of care robots, there has been an increasing need to understand the social value of care robots. This study attempted to identify the social values of care robots by conducting focus group interviews (FGIs) with twenty-four care recipients and caregivers and by using analytic hierarchy processes (AHPs) with thirteen individuals with expertise in the care service and care robot industries. Our results show that the labor- and health-related benefits, the technology innovation, and the provision of essential care work have the highest importance among the criteria of care robots’ social values. The criteria that receive lowest priority are cost, the autonomy and needs of the care recipients, and the organizational innovation. Our study suggests that along with the private benefits and costs of care robots, their social values also need to be considered to improve the quality of care and to unlock the potential of the care robot industries.

## 1. Introduction

Providing access to affordable and high-quality care is a primary concern for many advanced countries, especially those experiencing the demographic shifts driven by low fertility rates and aging populations [1]. The demand for health and social care is expected to increase in an aging society as a greater proportion of the population requires care for chronic illnesses, along with long-term care from either family members or professionals. At the same time, it becomes challenging to secure a sufficient number of care workers with the appropriate skills due to the decline in working-age population [2,3].

One of the potential solutions to the problem of providing adequate support for the elderly and the disabled, who are socially disadvantaged in many cases, while reducing the economic and social burden caused by an aging population, is the adoption of care robots. A care robot refers to a robot created for the purpose of providing physical, cognitive, and emotional support to professionals and caregivers, as well as assisting the disabled/elderly with difficulties in daily living [4,5]. Care robots are expected to reduce inconveniences in inpatients’ daily lives and to create high-quality caring jobs through a reduction in caregivers’ physical burden and stress [1,6]. Therefore, the utilization of caring robots can potentially enhance the entire process of caregiving.

While previous studies on care robots have focused mainly on the productivity and workability of care robots [7,8], there has been a growing need to understand the social value of care robots. Social value measures the positive value that contributes to the public interest, the creation of quality jobs, social integration, and community development, as well as the safety, environmental conditions, and emotional consideration of socially disadvantaged people [9]. Considering that all members of society have the universal right to care and the need and the duty to share care responsibilities in an efficient manner, it is important to evaluate the social value of care robots beyond the individual rights and benefits of caring robots [2].

The main purpose of this study is to estimate the social value of caring robots. To this end, we conducted focus group interviews (FGIs) with representative caregivers and care recipients. Using the results of the FGIs, we classified the social values of care robots into four criteria: labor, health, innovation, and economic values. For each classification of the social values, we further divided each value into two or three subcategories. Next, we conducted an analytical hierarchy process (AHP) to quantitatively evaluate the relative importance of each classified social value.

The results show that, in terms of labor value, providing essential care appears to be the most important value, followed by competent caregiving and security in the workplace. The importance of health value is presented in the order of function/health/safety, independent living and autonomy and needs of care recipients. For innovative value, the importance of technological innovation is more than twice that of organizational innovation, and for economic value, the importance of labor- and health-related benefits is three times larger than the importance of development and management costs. Among the entire classifications of the social values, the top five most important social values are as follows: technological innovation, labor- and health-related benefits, providing essential care, independent living, and security in the workplace. To our knowledge, our study is one of the few studies that attempts to measure the social values of care robots. We adopted a novel approach to defining and classifying the social values of care robots and to determining the relative importance of each classified social value.

The remainder of the paper is organized as follows: Section 2 explains the methodology used to define and measure the social values of care robots, while Section 3 presents the results obtained from the focus group interviews and our analytic hierarchy process. Section 4 discusses the broad interpretation of our results, and Section 5 concludes our argument.

## 2. Methods

This study employs a mixed methods design to identify the social values of care robots. Focus group interviews (FGIs), as a qualitative research method, and an analytic hierarchy process (AHP), as a quantitative research method, were jointly conducted to derive the core values of care robots (Figure 1).

### 2.1. Focus Group Interview

#### 2.1.1. Participants

Based upon the literature reviews, we specified the criteria for selecting the participants [2,3]. Nine care recipients, including five elderly subjects and four individuals with disabilities, and fifteen caregivers participated in the FGIs. Three out of the four individuals with disabilities were bedridden. Face-to-face individual interviews were conducted with the four individuals with disabilities. The caregivers were staff currently engaged in work at welfare centers for the elderly or for individuals with disabilities. Six students from the departments of nursing, social work, and mechanical engineering were included because new creative opinions regarding working with care robots or the invention of care robots in the near future are needed.

#### 2.1.2. Procedure

The FGIs were conducted from 12 to 24 January 2022. Our research team made short video materials to introduce care robots, such as a transfer-assist robot, a pressure ulcer-prevention robot, a meal-assist robot, and an excretion-assist robot, as the participants were not familiar with these robots as they are not popularly in use. After showing the 10 min video presentation about the operation of care robots, the researchers conducted interviews, including the following questions in Korean: are there any specific changes or impacts which stem from the emergence of care robots? What impact or changes do you think care robots will bring to caregivers, care recipients, or our society? Let us talk about the values, effects, utility, or good aspects of care robots for caregivers, care recipients, and our society. Do you think using a care robot is meaningful to you or to society? The interviews were recorded with the informed consent of the participants. Audio recordings of the interviews were transcribed by two research assistants.

#### 2.1.3. Analysis

A thematic analysis identifying, summarizing, and searching for repetitive patterns in the interviews’ contents was performed through transcription [10]. The distinct patterns were drawn from the comparison of the FGI contents, which was mainly led by two researchers (KP and JL). The final themes and keywords were extracted and selected with the help of a qualitative research expert (JA).

### 2.2. Analytic Hierarchy Process

#### 2.2.1. Participants

We created potential participant pools, which can mainly be divided into three groups: the professionals and researchers that have sufficient knowledge and experience in care robots; the seasoned staff that work in care facilities; and the government officials that actively design policies and regulations for care robots in their related fields. We then recruited participants from the pools, while maintaining a balance between the number of participants of each group, and we eventually finalized thirteen participants for the AHP procedure.

#### 2.2.2. Procedure

The AHP was conducted from 1 to 31 August 2022. We structured the AHP hierarchy based upon the pre-defined social values of care robots, primarily drawn from the previous FGIs. Table 1 shows the hierarchy of the goals, criteria, and alternatives at each level. At level 4, each social value means the following: (1) competent caregiving represents the value of a care robot that enables caregivers to provide excellent, professional, and high-level care for complex care tasks. (2) Providing essential care is the value of providing essential, but often simple and repetitive, care, such as bedside nursing. (3) Security in the workplace is the value of protecting caregivers from physical and emotional violence and workplace accidents. (4) Autonomy and need constitute the value of accommodating the autonomy, needs, and expectations of the care receiver. (5) Independent living is the value of the care robots that enable the care receiver to independently perform daily activities and self-care. (6) Function, health, and safety constitute the value of leading care receivers to consume an appropriate amount of food, to maintain safe movement and balance, and to improve various body functions. (7) Technological innovation is the value related to applying the new technology of care robots, including minimal weight and volume, uniformity, movement speed, and optimal design. (8) Organizational innovation is the value of care robots contributing to the ethical and social work environment, stakeholder management, sustainable thinking, and cooperative relationships. (9) Labor- and health-related benefits constitute the value related to the reduction in the physical and mental burden for caring, the creation of new jobs, the reduction in caring time, and the improvement in health status. (10) Robot development and management costs constitute the value which is related to the money spent on the development, production, and management of care robots through commercialization, new business opportunities, and the mass production of care robots.

We then finalized the participants and informed them about the goals and objectives of the research, as well as the protocols and procedures of the AHP. The team distributed the survey and collected the responses via email.

#### 2.2.3. Statistical Analysis

Based upon the initial responses, we primarily focused on confirming the consistency of the responses by checking their consistency ratios (CRs), which stemmed from a set of pairwise comparisons. As those responses whose CRs are under 0.2 are commonly acceptable in assessing attribute importance [11], we encouraged some participants whose responses might not satisfy the critical values to revise their responses. We eventually derived the local and global weights of each social value, the local scores of each alternative, and the global priorities of the social values of care robots based upon the revised and validated responses.

### 2.3. Ethical Considerations

This study was approved by the institutional review board of the author’s university (GWNUIRB-2021-36). The author and the research assistant provided the information about the purpose and procedures of the study to the participants, ensured the anonymity of the participants’ personal information, and used a separate study ID for each participant. Before the participants provided written consent, they were informed that there was no penalty for not participating and that they could withdraw at any time during the study.

## 3. Results

### 3.1. Focus Group Interviews

The results from the FGIs are summarized in Table 2. The following keywords emerged from the FGIs: care burden, hard work, employment, new job, abuse, safety concerns, wages, management, immediacy, 24 h continuity, sense of isolation, isolation, self-confidence, sense of accomplishment, less humane but less burdensome, less uncomfortable and less sorry, accuracy, standardization, belief, personalization, autonomy, legal/ethical issues, social agreement, care costs, additional costs, rental, and so on. These keywords were grouped into three themes, namely social values, innovative values, and economic values. Labor values and health values belong to the social values as sub-categories.

### 3.2. Analytic Hierarchy Process

Table 3 shows the local and global weights of the social values of care robots at each level. The local weight was calculated by the normalized geometric mean of a row in the pairwise comparison matrix; it reveals the relative importance of social values in the same category, i.e., the upper-level criterion. For instance, with regard to labor value, providing essential care (0.378) appeared to be the most important value, followed by competent caregiving (0.331) and security in the workplace (0.291). By comparison, global weight is the product of the local weights at all the levels and thus reveals the relative importance of the social values at the same hierarchical level, level 4. For instance, we could conclude that the importance of technological innovation (0.187) was two times larger than the importance of providing essential care (0.093).

Table 4 shows the global weights, local scores, and weighted scores of the social values of care robots at level 4. The local score was calculated by the normalized geometric mean of a row in the pairwise comparison matrix, and it reveals the relative importance of two distinct alternatives, i.e., promoting the development of care robots vs. stopping the development of care robots in terms of the social value at level 4. Additionally, as a result of the evaluation, the sum of the products of the global weights and local scores was reported; see Table 4. The results clearly show that promoting the development of care robots (0.828) was superior to its alternative, which was stopping the development of care robots (0.172). Finally, we derive the global priority of each social value, as drawn from the weighted scores at level 4. Among all the classifications of social values, the top five most important social values were as follows: technology innovation (19.68%), economic benefits (19.39%), provision of essential care labor (9.83%), independent living (8.59%), and security in the work environment (8.31%).

All reported local and global weights are the arithmetic means of local and global weights that stem from each individual participant’s responses.

## 4. Discussion

In the present study, labor/health values were suggested to be important sub-categories of the social values for care robots. The previous studies reported that care robots increased mobility, physical function, safety, and self-efficacy of care [12,13,14,15,16,17]. However, it is not sufficient for care robots to accommodate user preferences, needs, or expectations [3]. Therefore, we require the development of care robots and care services using care robots to meet the care recipients’ demands in order to increase their health values; this was apparent in the care recipient group of the FGIs carried out in this paper.

In terms of labor values, care robots could aid caregivers by providing positive physical or psychological tools, such as the by the prevention musculoskeletal diseases or a decreased care burden or stress [3,6,18]. As a result of the AHP, performing simple and repetitive tasks was one of the highest elements of the social values of the care robots. This reflects the expectation that performing simple and repetitive tasks would be the main roles for care robots instead of human caregivers. Basic care, such as assisting meal intake, getting rid of excrement, or cleansing the body, is very simple, repetitive, and trivial work with low value; however, these are essential processes in human life. Therefore, simple and repetitive tasks performed by care robots should be considered when developing a care service or a new function of care robots to maintain labor values.

Recently, the International Labor Organization (ILO) introduced the concepts of ‘Decent Work’ and ‘Decent Working Time for Nursing Personnel‘ [19]. A few previous studies have also suggested various conditions of decent work in nursing, such as a decent wage, a decent working time, stable employment, social security, a job training system, and a decent working environment, including the prevention of violence and the protection of human rights [20,21]. The ILO and previous studies suggest that efforts to achieve ‘Decent Work’ in the caring field will contribute to increasing the quality of care and resolve the care worker shortage issue. The purpose of ‘Decent Work’ is same as the purpose of developing care robots in terms of providing quality of care and a decreased care burden. In current robot technology, the level of development of care robots is not yet established enough to support people without a human caregiver. Most people should receive a care service with a care robot and a human caregiver together. Eventually, it is expected that care robots with the right technology will be able perform care services alone. However, in the near future this remains impossible because many aspects of this approach are yet to be resolved, such as the development of the technology and the reforming of the law and societal systems for the use of robot caregivers. This is why it is necessary to include values from human caregivers while identifying the social values of care robots.

Innovative value is the next social value that we examined. The definition of innovation is the generation of ideas, the successful development of that idea into a useable concept and, finally, the successful application of that concept [22]. Innovation introduced by technology, such as new machines or equipment, new materials, and new processes to produce ideas and new products, is referred to as technology innovation [3,22,23,24]. On the other hand, social innovation, which involves the creation of new organizations, business practices, and ways of running organizations or new organizational behavior, is referred to as organizational innovation [23]. Previous studies have reported that the care robot industry has a high growth potential. In order to realize their potential as a new industry, care robots should be marketed with high usability in order to reflect the close relationship between the product and the user, as well as the link between technology innovation and the environment and social systems, as realized through organizational innovation [3,25]. Organizational innovation is essential for the potential development of new industries [26]. However, studies regarding the innovative value of care robots have thus far been too focused on technology innovation. Therefore, further studies regarding organizational innovation values, such as environmental use [27], the voice of different stakeholders [28], ethical/societal issues, stigmatization [29], and cooperation in new relationships [30], are required.

Lastly, consistent investment and research into care robots led by governments and the private sector will contribute to the realization of the care robots’ economic benefits. Care workers usually receive lower wages than average workers, and these wages are often responsible for the high turnover of care workers [20,21]. If care robots can contribute to the increase in productivity among caregivers and a reduction in their physical and mental burdens, they can potentially lower care worker turnover through increased employment and wages. Additionally, in many countries, care work is provided by family members without payment. Considering that a large number of women leave the workforce to take care of their family members, the increased use of care robots, which can potentially reduce caring hours, might lead to higher female labor force participation and economic growth. Finally, if care robots reduce pressure ulcers and falls among patients [14], we can expect a considerable extension of patients’ lives, as well as a better working environment for caregivers.

## 5. Conclusions

Our study classifies the social values of care robots into labor, health, and innovative and economic values. Each criterion can be divided further into two to four subcategories. The labor values consist of the values that contribute to the enhancement of competent caregiving, providing essential care work, and improving the security in the workplace. The health values represent the values that contribute to meeting care recipients’ needs and autonomy, sustaining care recipients’ independent lives, and assisting the care receivers’ functioning, health, and safety. The innovative values are the values that contribute to technological innovation as well as organizational innovation. Lastly, the economic values include the cost of the production and management of care robots as well as the benefits related to health and labor. The AHP results show that certain social values such as the technological innovation and labor- and health-related benefits are more important than others, implying the importance of increasing investment or quality of care in related sectors.

There will certainly be other social values of care robots that we have not fully covered in this study. For example, as care robots handle more complicated tasks and become sophisticated, they are expected to take a more important role during a public health crisis such as the COVID-19 pandemic. Care robots can help doctors and nurses treat patients in a safe environment by disinfecting rooms and delivering food or prescriptions to the patients. It is also possible that care robots can assist in handling the extra work when there is a surge in patients [31]. In addition, the use of care robots can contribute to preventing the spread of infectious diseases by avoiding unnecessary human contact, thereby improving the health of patients. Lastly, some robots that are designed to communicate and share emotional sympathy with patients can contribute to the emotional stability of patients who must stay in an isolated environment.

It will be important to examine the several issues in the follow-up studies. First, it is worthwhile to explore the heterogenous effects of care robots based on the individual characteristics such as age, gender, or other personal traits. This is because certain social values can turn out to be higher in specific groups. For example, younger care receivers may have a stronger willingness to live a more independent and self-sufficient life; so, health values can be particularly prioritized among them. Next, there is a need to carefully analyze the safety of care robots and the responsibility issues in the case of an accident. Unless clear and proper guidelines are suggested for potential accidents, both caregivers and care receivers will be reluctant to use care robots. Lastly, since the costs of purchasing care robots can represent a substantial financial burden for many households, it will be important to discuss the efficient way to finance the cost of care robots, especially for those who are of socially disadvantaged groups. Along with these studies, it is important to pay attention to the complementarity or substitutability between human labor and care robots. Therefore, further studies that include a more comprehensive review will aid our understanding of care robots’ social values.

## Figures and Tables

**Figure 1 ijerph-19-16657-f001:**
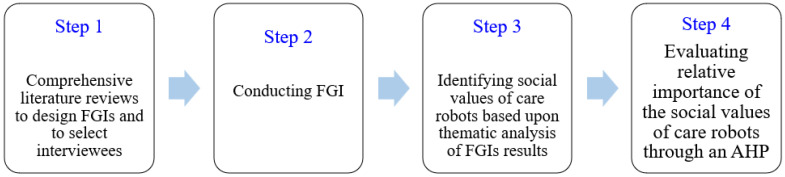
Research process of identifying the social values of care robots.

**Table 1 ijerph-19-16657-t001:** AHP hierarchy with the associated goals, criteria, and alternatives.

Level 1 Goal	Social Values of Care Robots			
Level 2 Criteria	Social Values		Innovative Values	Economic Values
Level 3 Criteria	Labor Values	Health Values	-	-
Level 4 Criteria	Competent Caregiving Providing Essential Care Security in Workplace	Autonomy and Needs Independent Living Function/Health/Safety	Technology Innovation Organizational Innovation	Development and Management Costs Labor- and Health-related Benefits
Level 5 Alternatives	Promote the development of care robots vs. stop the development of care robots			

**Table 2 ijerph-19-16657-t002:** The result of FGIs: keywords and themes regarding social values of care robots.

Example of Theme Sentence	Keywords	Theme
*“It takes a lot of strength to change the position of individuals with a disability.”* *“We should gently explain to people with dementia when we undress them and clean up their feces, even though they have a dementia and don’t understand the explanation. You can’t take their clothes off without any explanation.”* *“It’s hard to do chores repeatedly”, “I think it would be more convenient for robots to do chores.”* *“I think there will be a new job handling with machine, equipment like care robots in the future caring fields.”* *”Care workers are getting older. If using care robots becomes popular, I think that young people may choose caring jobs because older care workers are unlikely to want to operate care robots. The labor environment in caring job field will improve and the average wages of care worker will increase, while the care burden will decrease.”*	Caregiving intensity; care burden; competent caregiving; essential care; employment	Social values; labor values
*“I may have less contact with people, if I use care robots. I would feel less burdensome, less shameful, or less sorry to my robots caregiver.”* *“If I use robots caregiver, it would be good for me to be able to ask something anytime without feeling sorry.”* *“It is important for inventors to understand users’ needs and to produce robots that meet them.”* *“Doing something by myself is always good for me.”* *“I think that if I eat or deal with my feces by myself, my self-esteem will be maintained or enhanced.”* *“If I use a meal caring robots, my nutritional status will improve.”* *“I have an experience using a broke electric bed. It was a terrible experience.”*	Autonomy and needs; independent living; function/health/safety	Social values; health values
*“The robot can always come right away if I call it.”* *“Each person with a disability has his/her different needs, so customized robots are needed.”* *“There is a need for legislation and standardized practices for the use of robots.”* *“Standardized care will be provided if we use care robots.”* *“We need the consensus of the people in our society for care robots to become popular.”*	Technology innovation; organizational innovation	Innovative values
*“Mass production of caring robots will reduce costs.”* *“The cost of caregivers is currently higher than the cost of medical treatment.”* *“I have an experience for caregiver of my son to give extra pay. However, if I use care robots, I expect to save money.”* *“It may not be possible that national health insurance program covers patients and professionals against all the claims that can arise when using care robots.”*	Development and management costs; labor- and health-related benefits	Economic values

**Table 3 ijerph-19-16657-t003:** Local and global weights of social values of care robots.

Level 2 Criteria	Local Weights (A)	Level 3 Criteria	Local Weights (B)	Level 4 Criteria	Local Weights (C)	Global Weights (D): (A) × (B) × (C)
Social Values	0.484	Labor Values	0.531	Competent Caregiving	0.331	0.083
				Providing Essential Care	0.378	0.093
				Security in the Workplace	0.291	0.080
		Health Values	0.469	Autonomy and Needs	0.302	0.065
				Independent Living	0.345	0.083
				Function/Health/Safety	0.353	0.079
Innovative Values	0.255	-	-	Technology Innovation	0.699	0.187
				Organizational Innovation	0.301	0.068
Economic Values	0.262	-	-	Labor- and Health-related Benefits	0.748	0.196
				Development and Management Costs	0.252	0.066

**Table 4 ijerph-19-16657-t004:** Evaluation of alternatives and priorities of social values of care robots.

Level 4 Criteria	Global Weights (D)	Rank	Local Scores (E)	Rank	Global Priorities (G): (D) × (E)/(F)	Rank
Competent Caregiving	0.083	4	0.766	9	7.69%	7
Providing Essential Care	0.093	3	0.873	1	9.83%	3
Security in Workplace	0.080	6	0.857	5	8.31%	5
Autonomy and Needs	0.065	10	0.728	10	5.73%	10
Independent Living	0.083	4	0.862	4	8.59%	4
Function/Health/Safety	0.079	7	0.866	3	8.28%	6
Technology Innovation	0.187	2	0.873	1	19.68%	1
Organizational Innovation	0.068	8	0.769	8	6.31%	8
Labor- and Health-related Benefits	0.196	1	0.821	6	19.39%	2
Development and Management Costs	0.066	9	0.778	7	6.18%	9
Sum	1.000				100.00%	
Evaluation	Promote the development of care robots (F): ∑(D) × (E)					0.828
	Stop the development of care robots: 1 − (F)					0.172

## Data Availability

Data available on request due to restrictions e.g., privacy or ethical.

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
