# Peer review of "Social Values of Care Robots"

_ijerph, 2022, doi:10.3390/ijerph192416657_

Round 1

Reviewer 1 Report

1- Please do not use the exact word of the title in the keywords.

2- You need more references in the passage, especially in the abstract section

3- You can use other approaches and compare the results with AHP. Please mention some of these methods in the abstract and compare AHP with at least one.

4- How did you select the participants and are the number of participants enough to prove the efficiency of your methodology?

5- What are the barriers to this study?

6- What is the future direction of this paper?

7- Some figures can be added to this paper to make the methodology and processes more clear to the readers.

8- There is no section in the methodology describing the AHP process.

9- Please add a processing chain to describe the steps of this study.

10- References are a bit old. You must use more updated references.

Author Response

Thank you for your comments! We have revised the manuscript based on the reviewer’s comments. We also highlighted the revised section in the main text (in blue).

We have revised the manuscript based on the reviewer’s comments. We also highlighted the revised section in the main text (in red and purple). While revising manuscript, authors realized that English re-editing was required. Therefore, two files were uploaded. One is showing tracking of English editing and revising and the other is clear file.

1. Please do not use the exact word of the title in the keywords.

Thanks for the suggestion. We have revised keywords from ‘Analytic hierarchy process; Care robots; Economic values; Innovative values; Social values’ to ‘Analytic hierarchy process; Care robots; Economic values; Innovative values; Labor values, Health values’

2. You need more references in the passage, especially in the abstract section

Thanks for the suggestion. However, we prefer to include references in introduction and discussion section because having them in abstract can be distracting for the readers. We instead put extra effort to make the entire section of our paper clear and cohesive.

3. You can use other approaches and compare the results with AHP. Please mention some of these methods in the abstract and compare AHP with at least one.

Thanks for the suggestion. To the best of our knowledge, there exist a few approaches that could complement the AHP results such as cost-benefit analysis, contingent valuation methods, or meta-analysis. We are currently conducting a separate research project that evaluates the cost and benefit of care robots, but this approach has limitation because it measures only the direct costs and benefits of the robots. Since social value includes broader and indirect benefits of care robots, it is difficult to compare the results of AHP with those of cost-benefit analysis. Unfortunately, we are also not able to conduct the contingent valuation methods due to a limited budget and time. Similarly, because of the scarcity of the studies that examine the quantitative socio-economic effects of nursing robots, it was difficult to conduct a meta-analysis simultaneously.

4. How did you select the participants and are the number of participants enough to prove the efficiency of your methodology?

Qualitative research methods such as focus group interview do not require the strict number of participants to meet the statistical condition. Therefore, we selected the participants and the number of participants reflecting the previous studies [8],[17]. We added this in the line 83.

5. What are the barriers to this study?

We extend our conclusion section from line 297 with the discussion on limitation and future perspectives of our study.

6. What is the future direction of this paper?

We extend our conclusion section from line 286 with the discussion on limitation and future perspectives of our study.

7. Some figures can be added to this paper to make the methodology and processes more clear to the readers.

Figure 1 was added in line 78.

8. There is no section in the methodology describing the AHP process.

We have added the AHP process from line 121 to 142.

9. Please add a processing chain to describe the steps of this study.

Thank you for the suggestion. We have added the processing chain for the steps of this study in line 78. Figure 1 was added in line 78.

10. References are a bit old. You must use more updated references.

Thank you for the suggestion. We have added recent references.

Reviewer 2 Report

Please add Table 1 in the text and adjust the subsequent tables' numbering in the text-flow as well as the numbering of the tables themselves.

Please unify throughout the entire paper "care giver", "caregiver", "care-giver" as well as the derivates.

Please expand the Conclusions.

Please see attached file for a few more local correction suggestions.

Author Response

Thank you for your comments! We have revised the manuscript based on the reviewer’s comments. We also highlighted the revised section in the main text (in blue).

We have revised the manuscript based on the reviewer’s comments. We also highlighted the revised section in the main text (in red and purple). While revising manuscript, authors realized that English re-editing was required. Therefore, two files were uploaded. One is showing tracking of English editing and revising and the other is clear file.

1. Please add Table 1 in the text and adjust the subsequent tables' numbering in the text-flow as well as the numbering of the tables themselves.

Thank you for the suggestion. We deleted the participants` information before 1st submission because the limitation of the number of words in this manuscript, but we didn’t recognize our mistake.  We didn`t add the information because we have already described the participants` information from line 8 to 90.

2.Please unify throughout the entire paper "care giver", "caregiver", "care-giver" as well as the derivates.

Thank you for the suggestion. We unified throughout the entire paper "care giver", "caregiver", "care-giver" to caregiver.

3.Please expand the Conclusions.

Yes, we have added some information and authors` opinion to expand the conclusion in line 286-311.

4.Please see attached file for a few more local correction suggestions. 1) Line 28 require. 2) Line 66 tech-. 3) Line 90 table 1. 4 ) Line 121 table 1. 5) Line 144 table 2. 6) Table 2. please add main verb. 7) Line 156 table 3. 8) Line 166 table 4. 9) Line 179 please delete the final full-stop and the space between the asterist and the prior full-stop. 10) Line 180 delete the space

1)  We have revised to ‘requires’ in line 27.

2)  We have revised to ‘technology’

3)  We deleted the participants` information before 1st submission because the limitation of the number of words in this manuscript.

4) Table 1 is correct

5) Table 2 is correct

6) We have added main verb ‘have.’.

7) Table 3 is correct

8) Table 4 is correct

9) Yes, we have deleted the final full-stop and the space

10) Yes, we have deleted the space.

11) Line 259

- Add some explanation related to the limitation of the study

- Outline in detail further potential/possible study/research necessities and perspectives

- Elaborate more on the socio-economic impact of care robots

As suggested, we elaborate each social value at level 4 in AHP hierarchy (including socio-economic impact of care robots) and extend our conclusion section from line 286.

Reviewer 3 Report

Please refer to the attached file for the comments.

Author Response

Thank you for your comments! We have revised the manuscript based on the reviewer’s comments. We also highlighted the revised section in the main text (in blue).

We have revised the manuscript based on the reviewer’s comments. We also highlighted the revised section in the main text (in red and purple). While revising manuscript, authors realized that English re-editing was required. Therefore, two files were uploaded. One is showing tracking of English editing and revising and the other is clear file.

1. I suggest including the citations for the first paragraph of the Introduction section (line nr. 25-32).

As suggested, a few references were presented in introduction section.

2. I suggest reporting the names (specify them in abbreviations) that conducted theme analysis (line nr. 105-108)

Yes, we added the initial of researchers and expert` name.

3. From line nr. 53-58 is related to the methodology part. I suggest reporting them in section 2

Thank you for the suggestion. As following the authors` guidelines in this journal, the summarize of the methodology were described in line 49-55.

4. From line nr. 59-68 is related to the results section. I suggest reporting them in section 3

Thank you for the suggestion. As following the authors` guidelines in this journal, the summarize of the results were described in line 56-68.

5. Table 1 does not show the summary of the participant's information (line nr. 90). Please include the table reporting with participants involved in the study.

Thank you for the suggestion. We deleted the participants` information before 1st submission because the limitation of the number of words in this manuscript, but we didn’t recognize our mistake.  We didn`t add the information because we have already described the participants` information from line 84 to 90.

6. In the methodology section, it was not reported the language used to conduct FGIs. I suggest reporting the language used in FGIs.

As suggested, we reported language (in Korean) used to conduct FGIs in line 98.

7. I suggest reporting in detail the future perspectives (regarding the use of care robots).

8. Limitations of the study were not reported. If any study limitations, please report them.

We extend our conclusion section from line 286 with the discussion on limitation and future perspectives of our study.
